# Modeling the effects of contact-tracing apps on the spread of the coronavirus disease: Mechanisms, conditions, and efficiency

**Asako Chiba** \*

Post-doctoral Fellow, Tokyo Foundation for Policy Research, Minato-ku, Tokyo, Japan

\* asakochiba01@gmail.com

## Abstract

This study simulates the spread of the coronavirus disease (COVID-19) using a detailed agent-based model and the census data of Japan to provide a comprehensive analysis of the effects of contact-tracing apps. The model assumes two types of response to the app notification: the notified individuals quarantine themselves (type-Q response) or they get tested (type-T response). The results reveal some crucial characteristics of the apps. First, type-Q response is successful in achieving containment; however, type-T response has a limited curve-flattening effect. Second, type-Q response performs better than type-T response because it involves quarantine of those who are infected but have not become infectious yet, and the current testing technology cannot detect the virus in these individuals. Third, if the download rate of the apps is extremely high, type-Q response can achieve virus containment with a small number of quarantined people and thereby high efficiency. Finally, given a fixed download rate, increasing the number of tests per day enhances the effectiveness of the apps, although the degree of improved effectiveness is not proportional to the change in the number of tests.

## 1 Introduction

Contact-tracing apps have gained traction as strong policy tools to prevent the spread of the coronavirus disease (COVID-19). This is because stay-at-home orders, including social distancing measures, inevitably force people to decrease their expenditures and labor supply, which leads to the long-term contraction of the economy. Although a stay-at-home order is considered effective in impeding the virus spread, many countries are reluctant to assent to the heavy economic downturn caused by the stagnation in consumption and production that would occur if a stay-at-home order were imposed for months. To regain the level of social activities existing before the COVID-19 pandemic, it is essential that only those who are assumed to be spreading the virus be isolated from society. Thus, testing-and-tracing strategies offer hope to policy makers since they help to target infected individuals and their contacts. In particular, contact-tracing apps are expected to work because they shorten the time required to trace the contact networks [1] and can be implemented widely and easily because of the increasing popularity of smartphones.

**Data Availability Statement:** Although the individual census data are de-identified, they cannot be shared publicly because they are owned by the Ministry of Internal Affairs and Communication. Users' sharing data is prohibited by the terms of use

(https://www.nstac.go.jp/services/2ji/tokumei_kiyaku.pdf, only in Japanese), which are issued by the National Statistics Center (NSTAC). Data are available from NSTAC (contact via nstac-info@nstac.go.jp) for researchers who meet the criteria for access to confidential data.

**Funding:** The author, Asako Chiba, received funding from the employer, Tokyo Foundation for Policy Research (https://www.tkfd.or.jp/en/). The funder had no role in study design, data collection and analysis, decision to publish, or preparation of the manuscript.

**Competing interests:** The author has declared that no competing interests exist.

Despite their expected effectiveness, little is known about the effects of contact-tracing apps. Numerical analyses of the effects of the apps show that apps largely reduce the daily incidents only when more than 56% of the population downloads them [2]. Similarly, it has been argued that the apps should be downloaded by a sufficiently large number of people to ensure the effective reproduction number below unity [3]. Additionally, shortening the delay in receiving test results and detecting contacts is crucial [4]. Although these studies provide quantitative support for the view that the apps could successfully flatten the epidemic curve, a qualitative analysis of the mechanisms behind this finding is absent. In particular, several fundamental questions are left unanswered, namely, why these apps drastically reduce number of infections, whether their effectiveness can be replicated under any conditions, and the extent to which these apps are effective compared with other policies, such as test-only policies, stay-at-home orders, and their combinations. Above all, the trade-offs between the merits and shortcomings involved in adopting these apps should be discussed because, even if they mitigate the large-scale spread of the virus, everyone who comes in contact with those diagnosed with COVID-19 are required to self-quarantine, regardless of their health status.

In this context, this study provides a detailed analysis focusing on the effects of the contact-tracing apps and the mechanisms behind them. These apps notify individuals who have been in contact with those diagnosed. There are two types of responses to the notification (for details, see Section 2): the notified individuals quarantine themselves (hereafter referred to as type-Q response) or they get tested (hereafter, type-T response). The existing literature implicitly assumes the apps in their analyses belong to the former category [1, 2]. However, the current study is novel in that the type-T response has been taken into consideration, as well. The analysis employs a detailed agent-based model, which is largely an extension of Covasim, [5] an open-source tool to simulate the virus spread.

In the present study, the spread of the virus is simulated by using the individual census data from Japan. The main findings are as follows:

1. The apps' effectiveness depends on the intended response to a positive contact alert: The results reveal that the type-T response has a much weaker curve-flattening effect than the type-Q response.

2. With the type-Q response, almost full containment can be achieved. This is mainly because the type-Q response designates quarantine for those who have caught the virus recently but have not become infectious yet. Thus, the difference arises not because the COVID-19 tests of the notified app users, conducted as part of the type-T response, have 70% sensitivity (which allows 30% of the symptomatic to slip through the tests) but rather because the type-Q response quarantines uninfected people. Since those who have been recently infected cannot be detected with current testing technology, it is reasonable to suppose the type-Q response prevents the spread of the virus by complementing the limited ability of testing.

3. When the apps are downloaded by a sufficiently large number of people (more than 80% of all smartphone users, which is equivalent to 52% of the total population) under the type-Q response, not only are the number of cases drastically reduced, but also only a small proportion of people are quarantined. This is because if the download rate is extremely low, only a small proportion of those who had contact with the diagnosed receive the alert and are quarantined. Although the apps have limited or almost no downward pressure on the number of infected, the low download rate keeps the number of quarantined people small. On the other hand, when the download rate is nearly 100%, almost everyone who had contact with the diagnosed is notified and quarantined. This almost-complete tracing scenario

restricts the virus spread mainly because the infected are quarantined at the early stage of illness. Thus, even if the number of quarantined people per diagnosed person is large, the low number of diagnosed people eventually leads to only a limited number of people being quarantined. In other words, there is a feedback effect between the number of quarantined and infected. If the download rate is in the intermediate range, the apps result in only a limited number of benefits with regard to both the prevention of virus spread and the decrease in the number of quarantined.

4. Given a fixed download rate, increasing the number of daily tests on symptomatic people tends to enhance the effectiveness of the apps, although the degree of enhancement is not proportional to the increase in the amount of testing.

This study not only reveals the characteristics of the apps but also provides a qualitative analysis using an agent-based model. Many studies, including the literature cited above, present the results of simulations but do not provide sufficiently detailed analysis on the apps' mechanisms. This is because the models tend to be so complicated that it is difficult to detect the main factors that generate the results. This is a crucial shortcoming of research using an agent-based model, as has been frequently pointed out [6]. This study overcomes this issue thanks to the relatively simple structure of the model: Although many detailed scenarios are introduced, the model, so far, focuses only on epidemics. By conducting counterfactual experiments in various parameter sets, the analysis reveals the mechanism behind the results, which enables an intuitive understanding of each scenario.

The following sections present the model, data, results and analysis, and finally, the conclusions and perspectives for future research.

## 2 Model

The analysis employs a detailed agent-based model mainly based on Covasim, an open-source model developed to simulate the virus spread [5]. The tool provides a basic framework to determine virus spread at home, school, workplaces, and other general contacts, when people's attributions are given. However, the model lacks several features regarding how outbreaks occur in reality. Contact-tracing apps cannot be directly implemented because the contacts are neither identified nor recorded in their model. In addition, neither vaccines nor variants are considered. Thus, the model has been extended in this study to analyze the impact of the apps in a more realistic environment. First, people's contacts have been modified to more accurately reflect their environment: specifically, contacts at shops, restaurants, and nursing homes have been introduced to illustrate the contacts among service-industry workers and customers and among the elderly living in nursing homes and their caregivers. In addition, features observed in the outbreak have been reflected in the model, namely, super-spreading environments that tend to arise with a certain probability and people with severe or critical illnesses who are automatically isolated until they recover. Most importantly, contact-tracing apps that record past contacts have been included in the present model, as have other environmental factors such as vaccines and variants.

Python codes used in the simulation are shared in http://dx.doi.org/10.17504/protocols.io.bwaepabe.

### 2.1 Transmission through contacts

Interventions are defined as policy tools aimed to prevent virus expansion: they include apps, tests, quarantine, isolation, vaccines, and social distancing. The dynamics under no interventions are described in the model as follows: On the first day, COVID-19 is brought into a

**Table 1. Definition of status.**

| State | Definition | Detected by tests | Infectious | Infected | Symptomatic |
|---|---|---|---|---|---|
| Uninfected | Not infected | - | - | - | - |
| Noninfectious | Infected but not infectious yet | No | No | Yes | No |
| Presymptomatic | Infectious but not symptomatic yet | Yes | Yes | Yes | No |
| Moderate | Symptomatic but not in need of hospitalization | Yes | Yes | Yes | Yes |
| Severe | In need for hospitalization | Yes | Yes | Yes | Yes |
| Critical | In need for intensive care | Yes | Yes | Yes | Yes |

hypothetical society, and a person becomes infected. This is followed by daily transmission of the virus through contacts, assuming that the community and service-industry contacts are shuffled every day.

The probability of a susceptible person becoming infected after contact with an infectious person depends on the overall infectiousness common to all contacts, the place where they meet, the relative susceptibility of the former, and the relative transmissibility of the latter. The setting in which they meet (referred to as a layer) is assumed to be one of the determinants of the probability of transmission because the frequency and duration of the meeting depends on the type of contact. For instance, if the two people are linked in the family layer, they spend a longer time together than for the other types of contacts, such as workplaces and schools. Thus, the probability that an infected person transmits the virus to their contacts in the family layer is assumed to be higher than that in the other layers.

In addition, the elderly are more susceptible than the young (see Tables 1 and 2), and the transmissibility of the virus in the early stage of illness is twice as high as that in the ensuing period: The early stage refers to the first N days after becoming infected, with N defined as the minimum between the first four days or one-third of the duration of the illness [7]. The duration of the illness for each infected person is determined by how the disease progresses, including when and to what degree their condition worsens. The timing is determined at the point they become infected, based on the distribution of duration and transition rate presented in Table 3.

The overall infectiousness of the virus is calibrated by targeting the speed at which the number of diagnosed people increased in Japan from January to February in 2020, when no specific policies were in place [8].

## 2.2 Transition of status

Once a person becomes infected, they are initially noninfectious and therefore asymptomatic [9]. Their condition worsens across phases that correspond to certain probabilities: They might become asymptomatic infectious, symptomatic, severely ill, critical, or die, or recover from each stage (see Table 1 for the definition of each stage).

The transition probabilities are computed based on age-specific data on infections in Japan using an interval of 10 years, which reveals a higher probability of worsening condition for the elderly than for the young. The transition probabilities of illness were computed primarily

**Table 2. Age-dependent susceptibility.**

| | ∼9 | 10∼ | 20∼ | 30∼ | 40∼ | 50∼ | 60∼ | 70∼ |
|---|---|---|---|---|---|---|---|---|
| Relative susceptibility | .34 | .67 | 1.00 | 1.00 | 1.00 | 1.00 | 1.24 | 1.47 |

**Table 3. Durations and probabilities of status transition.**

|  | Duration of transition (days) | Probability of transition | | | | | | | | |
|---|---|---|---|---|---|---|---|---|---|---|
|  |  | ~ 9 | 10 ~ | 20 ~ | 30 ~ | 40 ~ | 50 ~ | 60 ~ | 70 ~ | 80 ~ |
| (Worsen) |  |  |  |  |  |  |  |  |  |  |
| Not infectious ⇒ Presymptomatic | ~ LN(4.6, 4.8) | 1.000 | 1.000 | 1.000 | 1.000 | 1.000 | 1.000 | 1.000 | 1.000 | 1.000 |
| Presymptomatic ⇒ Moderate | ~ LN(1.0, 0.9) | 0.500 | 0.550 | 0.600 | 0.650 | 0.700 | 0.750 | 0.800 | 0.850 | 0.900 |
| Moderate ⇒ Severe | ~ LN(6.6, 4.9) | 0.000 | 0.000 | 0.000 | 0.155 | 0.151 | 0.198 | 0.365 | 0.360 | 0.408 |
| Severe ⇒ Critical | ~ LN(3.0, 7.4) | 0.000 | 0.000 | 0.000 | 0.029 | 0.029 | 0.147 | 0.368 | 0.491 | 0.490 |
| Critical ⇒ Death | ~ LN(6.2, 1.7) | 0.000 | 0.000 | 0.000 | 0.146 | 0.182 | 0.218 | 0.255 | 0.291 | 0.327 |
| (Recover) |  |  |  |  |  |  |  |  |  |  |
| Presymptomatic ⇒ Recovered | ~ LN(8.0, 2.0) | 0.500 | 0.450 | 0.400 | 0.350 | 0.300 | 0.250 | 0.200 | 0.150 | 0.100 |
| Moderate ⇒ Recovered | ~ LN(8.0, 2.0) | 1.000 | 1.000 | 1.000 | 0.845 | 0.849 | 0.802 | 0.635 | 0.640 | 0.592 |
| Severe ⇒ Recovered | ~ LN(14.0, 2.4) | 1.000 | 1.000 | 1.000 | 0.971 | 0.971 | 0.853 | 0.632 | 0.509 | 0.510 |
| Critical ⇒ Recovered | ~ LN(14.0, 2.4) | 1.000 | 1.000 | 1.000 | 0.854 | 0.818 | 0.782 | 0.745 | 0.709 | 0.673 |

using the number of people in each stage, as reported on June 10, 2020 [10]. The fact that the reported cases only include those confirmed positive and that a small fraction of symptomatic people could not get tested because of limited testing capacity suggests that the true number of infected people could be much higher than the reported cases. According to the report on anti-body-testing conducted in Tokyo from June 1 to 7, 2020, 0.10% of the population were anti-body positive, whereas the cumulative number of the confirmed positive as of May 30 accounts for 0.038% [11]. Thus, the model assumes that the true number of the infected in each age group is three-fold that of the reported cases.

In addition, the number of days it takes for the infected individual to go from one stage to the next follows a log-normal distribution with the moments exogenously set, which is also based on the real data (Table 3). In reality, people who are severely ill and critical are hospital-ized or cannot participate in social activities and therefore stay at home. Thus, it is assumed that they have no outside contacts, with contacts at their homes and nursing homes decreasing by 20% from the levels in normal times.

## 2.3 Super-spreading environments

A super-spreading environment refers to the phenomenon in which a small amount of cases accounts for a large amount of transmissions [12]. Moreover, it has been argued that the occurrence of this phenomenon depends on the environment rather than on the infected indi-viduals [13] and that the Pareto principle, namely that 80% of all consequences come from 20% of the causes, could apply in the case of COVID-19 super-spreading environments [14]. Therefore, it is reasonable to consider that transmission probability for 20% of contacts selected at random is 50 times as high as that for the remaining contacts in the simulations. In fact, super-spreading environments are one of the determinants of coronavirus spread. As described in Appendix B in S1 Appendix, introducing super-spreading environments into the model decreases peak cases by 40%-50%. This means that the simulations overestimate cases if super-spreading environments are not considered.

## 2.4 Apps, tests, quarantine, and isolation

When determining contact-tracing apps' effectiveness, it is important to consider the response the apps give to a COVID-19-positive notification. For example, those who are notified by the apps could stay at home, [2] whereas in Japan, the government requires them to get tested. To

clarify the relevance of the use of the apps, this analysis compares the difference between the effects of these two types of response, referring to the former as type-Q and the latter as type-T.

When contact-tracing apps are introduced, a certain fraction of people between 15 and 70 years old download the apps in the initial period: The reported usage rate for smartphones differs across age groups, with a relatively small fraction of people below 15 and over 70 years old having one [15]. For simplicity, the model assumes that only those between the ages 15 and 70, who account for 65.8% of the total population, have smartphones. Moreover, the term *download rate* denotes the ratio of the number of app users to the number of smartphone users.

In every period thereafter, COVID-19 tests are conducted on a daily basis, which implies that randomly selected symptomatic people are tested, and those who test positive are considered as diagnosed and, consequently, isolated. It is assumed that it takes a day for tested people to receive their results, since the reported lag is one day in many cases, with some cases of two or three days in Japan [16]. Isolation is defined as complete refrain from any type of contact until recovery, which describes hospitalization in reality. Sensitivity of the tests is set to 70%, which means that the tests can detect only 70% of the infected people who get tested. Those who have tested positive and have downloaded the apps register their diagnosis information as soon as they know the results. The app users are immediately notified if they had probable contact with the diagnosed individual within the last seven days. The action that should be followed by the alerted app users depends on the type of app. For the type-Q response, the alerted app users are indicated they should self-quarantine for 14 days after the notification. In contrast to isolation, people in quarantine are assumed to reduce their daily contact with people outside their homes by 90%, while their contact with their family members, or with those in nursing homes depending on the case, would be as per usual. With the type-T response, the alerted app users should get tested. If they test positive on the following day, they should commence self-isolation.

In these two scenarios, people are assumed to either quarantine or get tested depending on the response type. Further, all app users register the test results if they are diagnosed positive for COVID-19. In addition, the apps in the model are assumed to register all contacts in the past seven days, without any error. These strong assumptions do not hold in reality. That is, with regard to people's behavior, some app users may not register the test results or may feel reluctant to self-quarantine. Regarding the technology, reports of app errors suggest that they may fail to register a fraction of the contacts, or they may count two people sufficiently distanced and with no actual contact as a contact. Such phenomena are taken into consideration in the simulations, as shown in section 5.

## 2.5 Sequence of events

The intra-day events including the interventions as described take place in the following order: First, the community and service industry contacts are shuffled, and interventions such as testing, quarantine, and isolation are applied if necessary. Second, the susceptibility of the uninfected is computed based on their age and whether they are using any intervention like isolation or quarantine during the period. Third, the transmissibility of the infected is computed based on whether they are in the early stage of illness and whether they are following any intervention. Fourth, people come in contact with each other in each layer. The infected people probabilistically transmit the virus to those who are uninfected when they meet with others. Finally, all app users' health status is updated reflecting information such as whether they have been infected, their condition has worsened, or they have recovered.

## 3 Data

The analysis uses individual census data from Japan from 2015. Data were sampled as follows: From 125 million respondents, 25,000 were randomly selected. The records of those younger than 15 years and those who answered that they do not go to school were removed. The attributes include age, sex, job, employment status, and data on their family members. Employment status includes information on whether the respondent is employed, unemployed, educated, or none of these. Here, education refers to schools only and does not include kindergartens or nurseries; thus, school-age children are categorized as "none of these." In reality, about 60% of children younger than seven years attend kindergartens or nurseries, [17] where infection could spread. Therefore, for the analysis, a randomly selected 60% of school-age children are assumed to attend kindergartens or nurseries. Information on family members includes whether the respondents live in nursing homes or not. The analysis takes into account the community in the nursing homes, since the behavior of the elderly is thought to be one of the crucial factors that determines the severity of virus spread [18]. For employed respondents, the industry they work in and the size of the firm they work for are added as attributes based on their distributions, conditional for age and sex, which are obtained from the 2016 Economic Census for Business Activities.

For each respondent, a hypothetical family is created based on family member information. Each family member's attributes are assigned following their distributions, conditional on age and sex, which are obtained from the census data. This action, which creates approximately 50,000 additional hypothetical people, increases the number of middle-aged and young individuals who live with their families. This creates a distortion in the population's age distribution, that is, the estimated proportion of the elderly is smaller than that in reality. To adjust this bias, the number of the elderly who live alone is doubled. In total, the hypothetical society contains 75,614 people.

People in this society are assumed to have contacts through the networks in six places (see Table 4): Family networks are automatically formed when family members are created. Workplaces consist of the group of working people in the same industry and prefecture and who work for firms of the same size. Schools comprise a group of up to 25 children and at most two teachers in each prefecture, since the ratio of the number of teachers to students in elementary and junior high schools was about 7% [19]. Community contacts are the networks that link people randomly with an expected size of 10. Similarly, service-industry contacts are the link between service workers and groups of 20 randomly selected people (expected size) for each service worker. Nursing homes networks are constructed by grouping up to 20 people over 64

**Table 4. Methods used to create the networks for each layer.**

| Layer | Methods to construct networks based on the answers to the questions on family members | Average size (number of people) | Relative likelihood of transmission |
|---|---|---|---|
| Home | Constructed by the answers to the questions on families. | 3 | 50 |
| Workplace | For each prefecture, construct a group of individuals working for the same industry of a size that follows the firm-size distribution in the industry. | 5 | 5 |
| School | For each prefecture, construct a group of up to 25 educated individuals, adding 2 teachers in each group. | 25 | 5 |
| Community | Construct a group of individuals, randomly, of a size that follows a Poisson distribution with mean 10. | 10 | 1 |
| Service industry | Link each sale/customer service person with a group of randomly selected individuals of a size that follows a Poisson distribution with mean 20. | 21 | 5 |
| Nursing home | For each prefecture, construct a group of up to 20 elderly persons over the age of 60 years, adding up to 6 care workers in each group. | 25 | 50 |

years of age who live in care facilities and adding up to six care workers, for each prefecture, since the guidelines issued by the Ministry of Health, Labour and Welfare set the standard for the number of care workers in each nursing home to be one-third of the number of the residents.

## 4 Results

This section presents the simulation results. Each graph shows the average value of the results obtained in the simulations conducted 100 times. Each scenario assumes that a randomly selected person becomes infected on February 14, when the actual first positive individual was confirmed in Japan. When comparing the cumulative number of those who are infected and those who are quarantined, it is necessary that the number of days simulated is sufficiently large, such that the peak is attained and the virus outbreak ends long before the final day of the simulation in each scenario. In this analysis, it is set to 500 days. Tests and apps are assumed to start on day 33, and are assumed to be in place until the final day in the simulations.

### 4.1 Type-T response

Fig 1a–1c show the simulated time series data of the proportion of newly infected people in the population, under the conditions that daily tests are conducted on 30%, 50%, and 70% of randomly selected symptomatic people, respectively. The blue dashed line shows the result in the baseline scenario, where no interventions including tests or apps are introduced. The other solid lines show the results in the scenarios where 0% to 100% of people, aged between 15 and 70 years, download the apps with a type-T response: Tests are conducted on a daily basis, the diagnosed register their test results, and the alerted app users get tested. In any test probability, the number of infections decreases as the download rate increases: In the extreme case where the download rate is 100%, the peak value is about one-third of that in the baseline scenario, and the peak is delayed by one to two months. Although these figures show that type-T response flattens the curve, they also indicate the limitation of this type of response. Even if 70% of symptomatic individuals get tested and everyone in the target age group have downloaded the apps, no less than 0.03% of the population becomes newly infected at the peak. Further, the peak value of the sum of severe and critical patients, who need hospitalization, accounts for 0.06% of the population. This would overwhelm hospitals, considering that the number of beds prepared for COVID-19 patients as of April in 2020 in Japan was 32,000, which is equivalent to 0.03% of the population [20]. Additionally, social losses would be heavy, such as a substantial increase in government subsidies to the private sector to compensate for losses incurred by mandatory store closures [21] and economic stagnation. The amount of subsidies is high if the peak infection is high because the government then has to impose stringent and prolonged stay-at-home orders. In other words, a high peak not only harms people's health but also requires large-scale government spending. Thus, the type-T response's limitations in preventing virus spread are substantial.

### 4.2 Type-Q response

Similarly, Fig 2a–2c show the simulated time series data of the proportion of newly infected people in the population, under the condition that daily tests are conducted on 30%, 50%, and 70% of randomly selected symptomatic people, respectively. The result is compared to the baseline scenario and when the apps are introduced with the download rates varying from 0% to 100%; type-Q response is applied. Compared with the type-T response, the type-Q response has a substantial effect on flattening the curve: If the probability of daily tests is 70% and the download rate is 90%, the maximum number of newly infected individuals after the

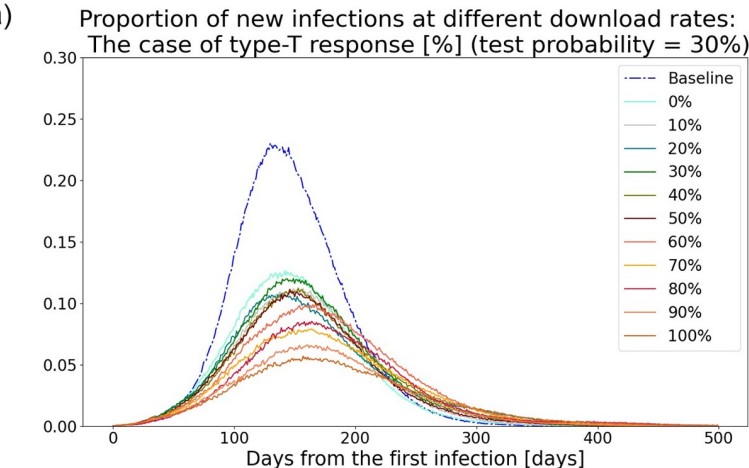

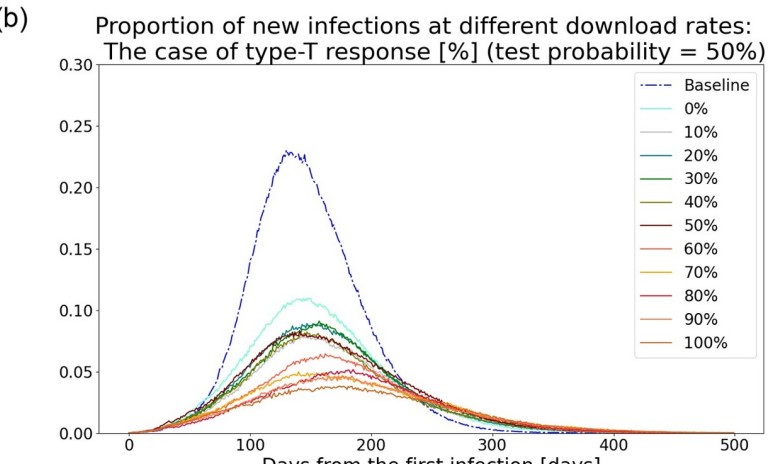

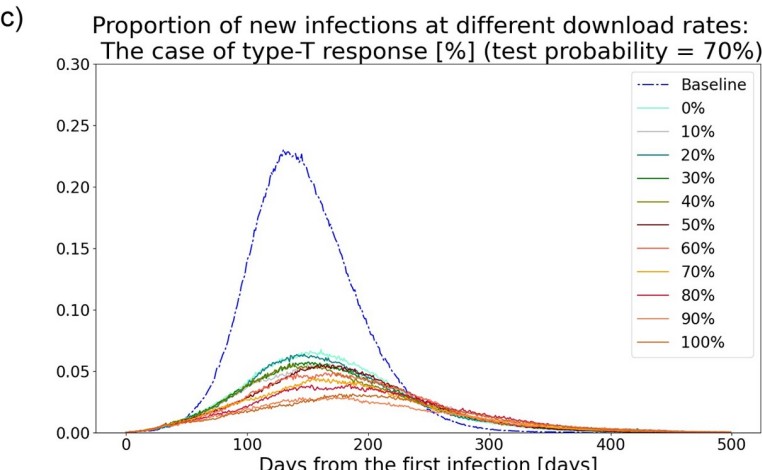

**Fig 1. Proportion of new infections in the population under conditions of daily tests on randomly selected symptomatic people: Type-T response.** (a) Scenario of 30% test probability. (b) Scenario of 50% test probability. (c) Scenario of 70% test probability.

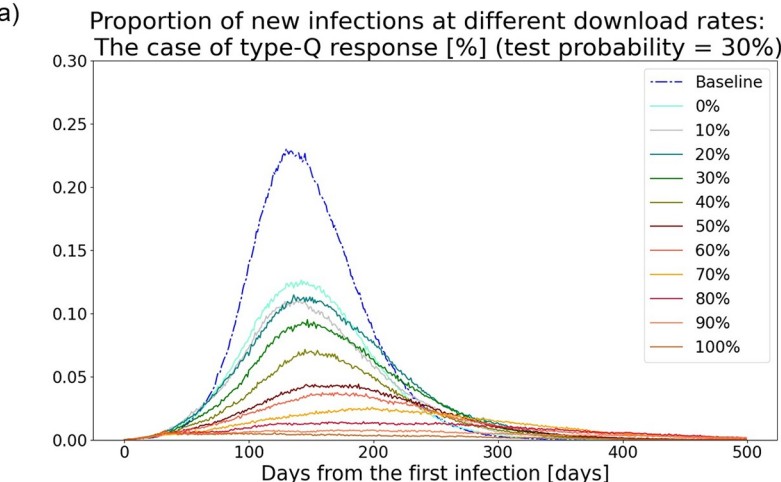

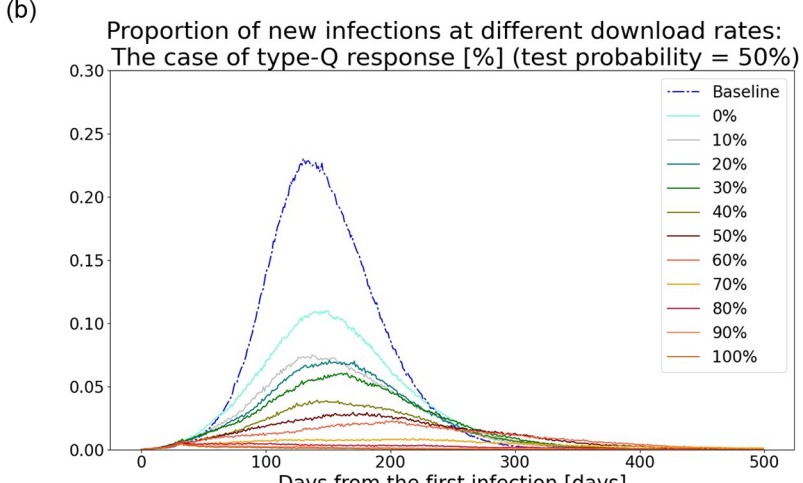

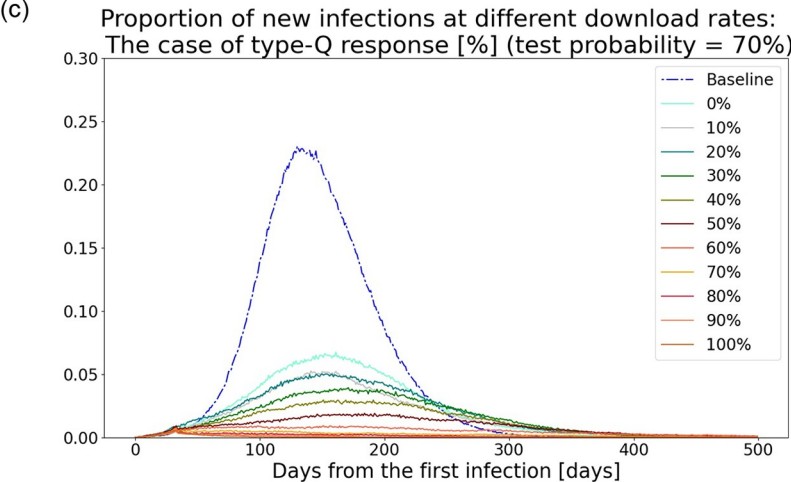

**Fig 2. Proportion of new infections in the population under conditions of daily tests on randomly selected symptomatic people: Type-Q response.** (a) Scenario of 30% test probability. (b) Scenario of 50% test probability. (c) Scenario of 70% test probability.

**Table 5. Scenarios to identify the main driver of the effectiveness that type-Q response has.**

| Scenario | Contents | Daily tests on the randomly selected symptomatic | Detectable status of contacts | Sensitivity and days required for testing contacts | Difference from Type-T scenario |
|---|---|---|---|---|---|
| Baseline | Without any policy | No | - | - | - |
| Type-T | Daily tests on the randomly selected symptomatic + type-T response | Yes | Infectious | 70%, 1 | - |
| TC1 | Counterfactual (variant of Type-T scenario) | Yes | Infectious | 100%, 0 | ①False-negative |
| TC2 | Counterfactual (variant of Type-T scenario) | Yes | Infected | 100%, 0 | ①False-negative ②Noninfectious |
| Type-Q | Daily tests on the randomly selected symptomatic + type-Q response | Yes | - | - | ①False-negative ②Noninfectious ③Stay-at-home |

introduction of the apps is two out of 75,000. In fact, even in the case where the download rate is 60% with the same probability of tests, or the download rate is 90% and the probability of tests decreases to 30%, the curve is almost flat. Although these extremely flattened curves appear only if the download rate is sufficiently high, these figures indicate the remarkable effectiveness of the type-Q response.

A natural question arises as to what drives the difference between the curve-flattening effects of type-T and type-Q responses. There are three possible factors: *test sensitivity effect*, *test target effect*, and *stay-at-home effect*. The first and the second are related to the limitations of testing. For example, 70% sensitivity means that 30% of symptomatic people are undetected by the tests and possibly spread the virus. In addition, tests can detect only those who are symptomatic [22]. Thus, noninfectious people, those who have caught the virus recently and are not infectious yet, cannot be detected. These factors may lead to the relative inferiority of the type-T response, which applies tests, compared to the type-Q response, which applies quarantine instead. Finally, the stay-at-home effect refers to the fact that the type-Q response does not distinguish app users' health status: The apps indicate all users who had contact with the diagnosed should self-quarantine, including those who are uninfected. For the alerted app users who are uninfected, a quarantine has a similar effect as a stay-at-home order, which protects them from being infected.

To identify the main driver, counterfactual experiments were conducted, assuming two hypothetical scenarios (Table 5). In one scenario, TC1, 100% sensitivity of the tests conducted for the alerted app users is assumed. In the second scenario, TC2, these tests can detect not only the symptomatic but also the noninfectious. By comparing the results in TC1 and those from the type-T response, one can estimate the test sensitivity effect. Similarly, the difference between the TC2 and TC1 results measures the magnitude of the test target effect, and the difference between the type-Q response and TC2 results measures the magnitude of the stay-at-home effect.

Fig 3a–3c show the proportion of the cumulative number of newly infected people in the population under the condition that daily tests are conducted on 30%, 50%, and 70% of randomly selected symptomatic people and compares the result in the different scenarios—type-T response, TC1, TC2, and type-Q response. The download rate is varied from 0% to 100%. One can observe that there are fewer cumulative infections in TC1 than in the type-T response. However, the difference is not substantially large enough to account for the large gap in the results with the type-T and type-Q responses. Rather, the TC2 results are similar to those of the type-Q response. This observation leads to the identification of the main factor of the curve-flattening effect of the type-Q response: It prevents virus outbreak because the

(a)
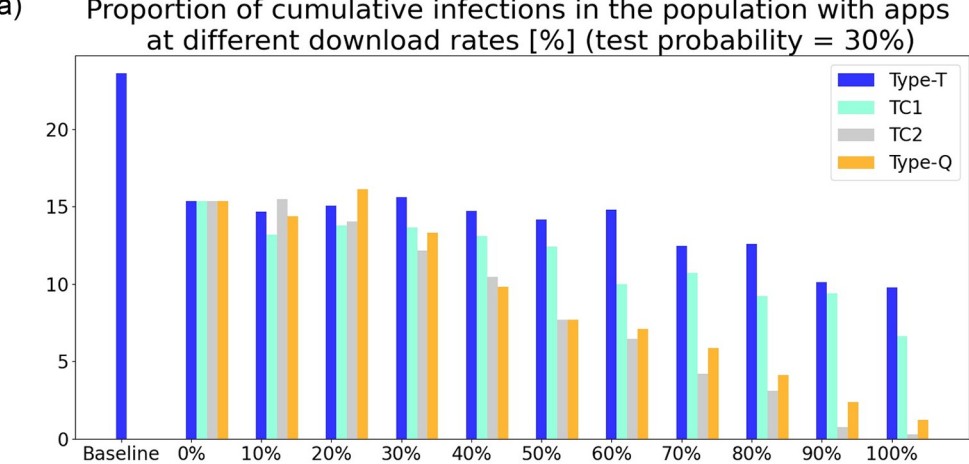

(b)
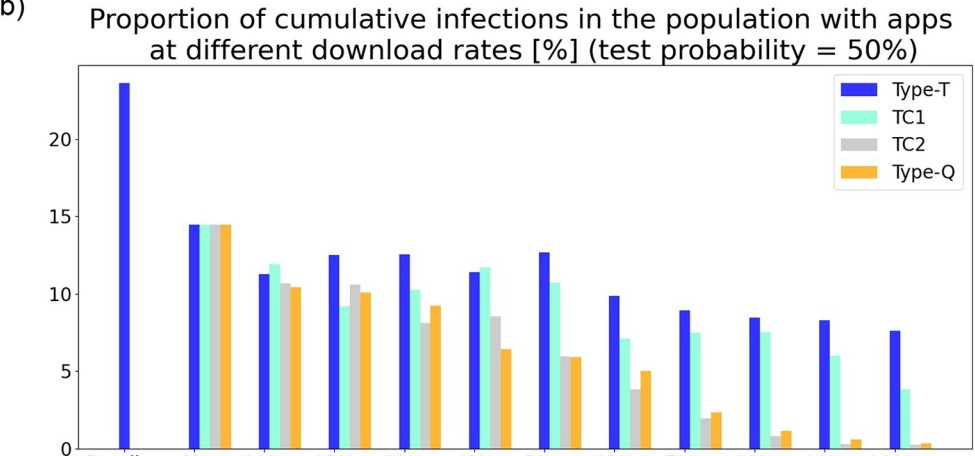

(c)
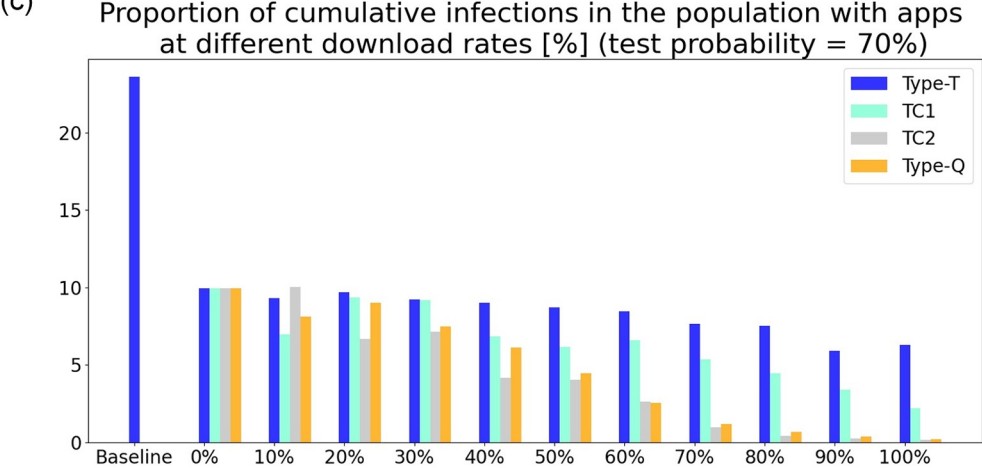

**Fig 3. Proportion of cumulative number of infections in the population under different scenarios.** (a) Scenario of 30% test probability. (b) Scenario of 50% test probability. (c) Scenario of 70% test probability.

noninfectious quarantine. The result is consistent with the literature that reports that the silent transmission of the virus from presymptomatic or asymptomatic people might be the main source of outbreaks ([23]; for a literature review, see [24]). From a practical point of view, the result also suggests the relative effectiveness of semi-targeted quarantine over testing. In reality, tests cannot detect those who are in the early stage of illness, although their behavior is the key to determine the speed and degree of transmission of the virus. Thus, the type-Q response complements the limitation of testing by quarantining all app users who had been in contact with the diagnosed, including those who have just caught the virus.

## 4.3 Effectiveness of the type-Q response

Thus far, the apps have been evaluated according to their curve-flattening effect, that is, how many people have been able to stay healthy. However, the main aim of policy is to prevent the spread of the virus at low social costs. Thus, *efficiency* matters. Since the model does not contain any economic parameters, first, the number of quarantined individuals and the number of infections were compared. Hereafter, only the type-Q response is analyzed.

Fig 4a–4c show the relation between quarantine and infection under the condition that daily tests are conducted on 30%, 50%, and 70% of randomly selected symptomatic people, respectively. In each figure, the proportion of the cumulative number of quarantined people in the population is shown on the x-axis, and the proportion of the cumulative number of infections in the population is shown on the y-axis. The scatter plots depict the results as the download rate of apps is varied between 0% and 100% and type-Q response is applied. For comparison, the result in the baseline scenario is also plotted. The apps are more effective if the point is located in the lower-left quadrant of the graph because it means that the apps prevent the spread of the virus with a small fraction of people getting quarantined. The plots illustrate scenarios in which symptomatic people are tested with a probability 30%, 50%, and 70%.

Regardless of the probability of testing symptomatic people, the apps can contribute to the quarantining of a relatively large number of people if the download rate is around 40% or 50%. The reason is as follows: If the download rate is extremely low, only a small fraction of those who had contact with the diagnosed receive the alert and can then quarantine. Although the apps have limited or almost no downward pressure on the number of infected people, the low download rate keeps the number of quarantined people small. On the other hand, when the download rate is nearly 100%, almost all people who had contact with the diagnosed are notified and can then quarantine. This almost-complete tracing impedes virus transmission mainly because the infected are quarantined at an early stage of their illness, as explained in the previous section. Thus, even if the number of quarantined per diagnosed person is large, the low number of diagnosed people eventually leads to only a limited number of quarantined people. In other words, there is a feedback effect between the number of quarantined and infected people.

In this context, what happens if the download rate is in the intermediate range? The apps would neither prevent the spread of the virus nor decrease the probability of tracing. As a result, both the number of infected and quarantined people remain high. This mechanism can be also observed in the time series data. Fig 5 shows the proportion of newly quarantined people in the population when the daily tests on symptomatic people are conducted with a probability of 30%, and the download rate of the apps is 10%, 60%, or 100%. With the introduction of the apps, if the download rate is 100%, the number of quarantined people spikes to 0.7% of the total population, which is nevertheless followed by a decreasing trend. When the download rate is 60%, the initial spike is lower than that in the case of a perfect download rate. However,

(a)

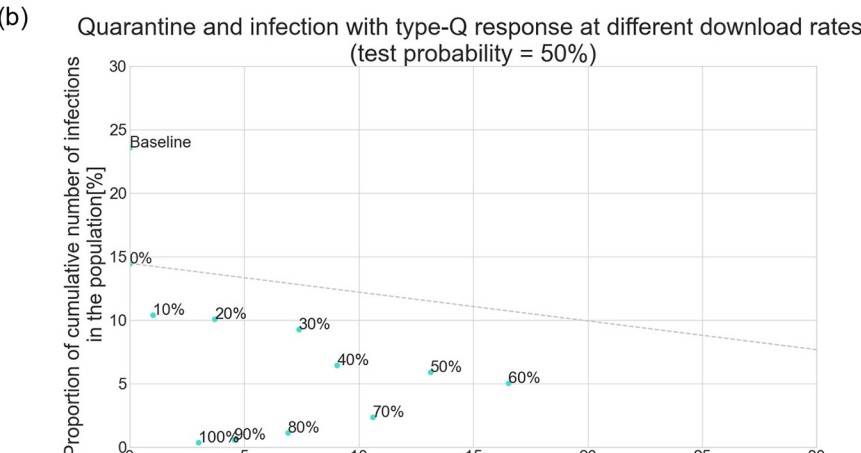

(b)

(c)

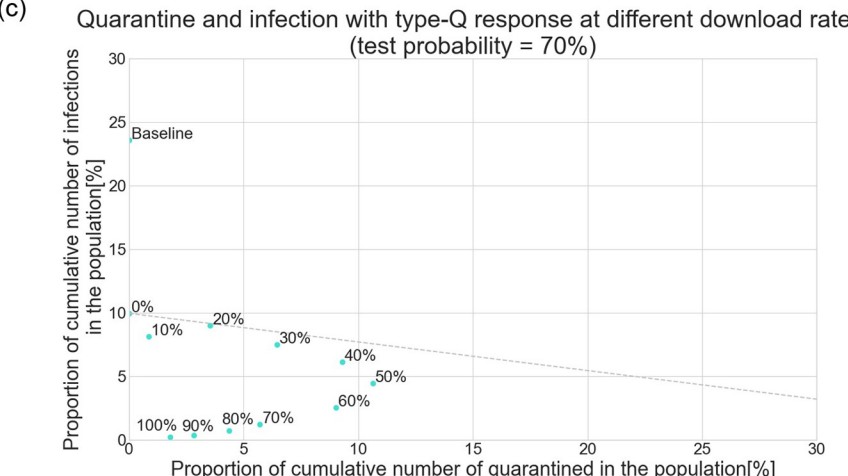

**Fig 4. Quarantine and infection when the download rate of apps is varied and type-Q response is applied.** (a) Scenario of 30% test probability. (b) Scenario of 50% test probability. (c) Scenario of 70% test probability.

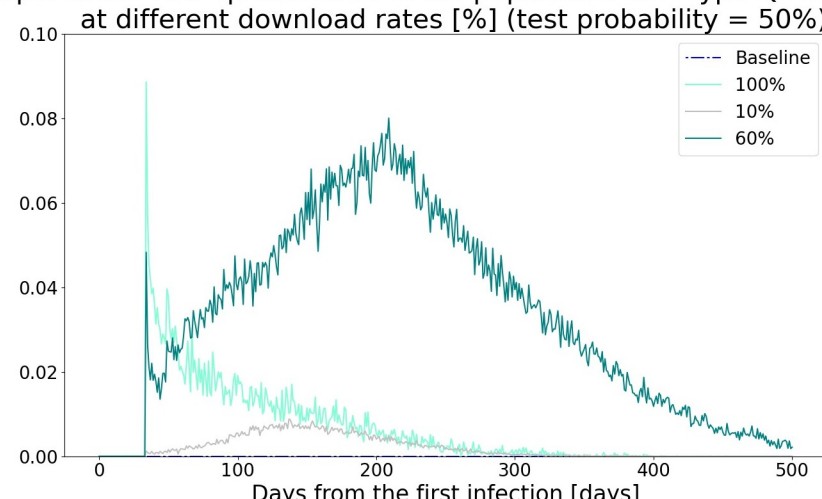

**Fig 5. Time series of the number of quarantined people at different download rates.**

the subsequent increase in the numbers shadows the spread of the virus. This leads to a substantially large number of people self-quarantining over time.

To analyze the efficiency more in-depth, a simple method to estimate social costs is applied: comparing the product of the number of quarantined people and the duration of quarantine to the product of the number of infected people and the duration of inactive time (see Appendix A in S1 Appendix for the computation of the duration of inactive time). Approximating social cost using this method is rationalized by the fact that people who are in quarantine or isolation cannot participate in economic activities. Thus, the social costs should increase as the number of isolated/quarantined people increases, and the duration of isolation/quarantine becomes longer.

The dashed line in Fig 4a–4c shows the download rate at which the apps perform better than the testing-only scenario: Its intercept is located at the point in the case where the download rate is 0%, and its slope is the ratio of the expected duration of quarantine (14 days) and that of isolation (conditional on being infected). If a point is located beyond this frontier, the use of apps creates a relatively large amount of inactive labor force and consumption compared to a policy wherein only daily tests are conducted and no apps are introduced. If 30% of the symptomatic get tested, the use of apps performs better than the test-only policy, but only if they are downloaded by more than 90% of people between the ages of 15 and 70 years. As the probability that the symptomatic get tested increases, using the apps to quarantine/isolate tends to mitigate the spread of the virus with a smaller number of people being quarantined. Although the increase in testing probability shifts the frontier downward, the effects are amplified if apps are introduced; thus, the degree of each point's shift to the bottom-left quadrant is larger than that of the frontier.

Overall, using the apps to quarantine/isolate performs better than testing-only policy regardless of the probability of testing the symptomatic and the download rate. As the download rate increases, the efficiency frontier shifts downward, shown by the line parallel to the dashed line and corresponds to each download rate across the plots, although the degree of shift is limited if the point moves to the bottom-right quadrant. Once the download rate reaches the turning point where the point starts to move to the bottom-left quadrant, the frontier moves downward at a larger degree. These results emphasize that the apps, from the

viewpoint of efficiency, should be downloaded by a sufficiently large number of people; otherwise, they result in a limited amount of efficiency gain, which appears as the downward shift of the intercept of the frontier. In addition, the results imply that, with a fixed download rate, increasing the number of daily tests of the symptomatic tends to enhance the effectiveness of the apps; however, the degree of enhancement is not proportional to the increase in the probability of testing. For instance, if the download rate is 90%, the effectiveness gain when testing probability increases from 50% to 70% is much smaller than when testing probability increases from 30% to 50%. Thus, app efficiency is not always improved by increasing testing probability.

## 5 Robustness

As mentioned in section 2, people and apps do not behave as they are expected to: Some app users may feel reluctant to register the positive test results, and others may not accept to self-quarantine even if they are alerted to having been in contact with a person confirmed positive for COVID-19. With regard to the apps, various errors have been reported (see, for example, [25]). Thus, the assumption in the main analysis that all app users will register test results and quarantine themselves when they receive an alert can be considered optimistic.

Fig 6a–6c depict the relationship between the cumulative numbers of quarantined and infected individuals under the assumption that the probability of registration, apps' perception rate, which is defined as the probability that the apps detect positive contacts, and probability of self-quarantining upon receiving the alert are varied between 0% to 100%, respectively. The daily tests are conducted on randomly selected symptomatic people with a probability of 50%, and the download rate is set to 90%.

Fig 6a and 6c exhibit the same pattern; if these rates decrease to 30%, the number of infections increases, and as a result, the number of quarantined people increases, thereby implying that the apps' performance has worsened. The results are different when the perception rates are imperfect, as shown in Fig 6b. Except for the case where the perception rate is 10%, the apps' effects on the number of infected and quarantined people do not depend on the perception rate. This is because the apps fail to catch a certain proportion of the contacts that are randomly selected from all the contacts in the past seven days. Those who are linked at the home, nursing home, school, and workplace layers have daily interaction, whereas the contacts in the other layers are shuffled every day. Thus, even if the apps' perception rate is low, such as 30%, the transmissions for the layers where contacts are fixed are likely to be traced. Moreover, in these layers, people tend to stay in close proximity for a long time, which is reflected in the high likelihood of virus transmission (Table 4).

To summarize, the number of the infected and thus the number of quarantined individuals are rarely affected by the apps' perception rate because key contacts are detected regardless of the apps' ability; when the probability of registration or of self-quarantine is less than 1, the number of effective app users is lower than the number of people who downloaded the app. Even if the download rate is as high as 90%, the small proportion of effective app users leads to the inefficiency problem observed in Fig 4. Thus, people's honest registration of their test results and following self-quarantining are crucial for the apps to contribute to efficiently reducing infection numbers.

## 6 Vaccines, variants, and social distancing

Thus far, the analysis assumes the simplest scenario in which interventions other than testing and apps are not implemented and only preexisting variants are transmitted. However, at the time of writing, vaccines have started to be distributed in many countries, and more

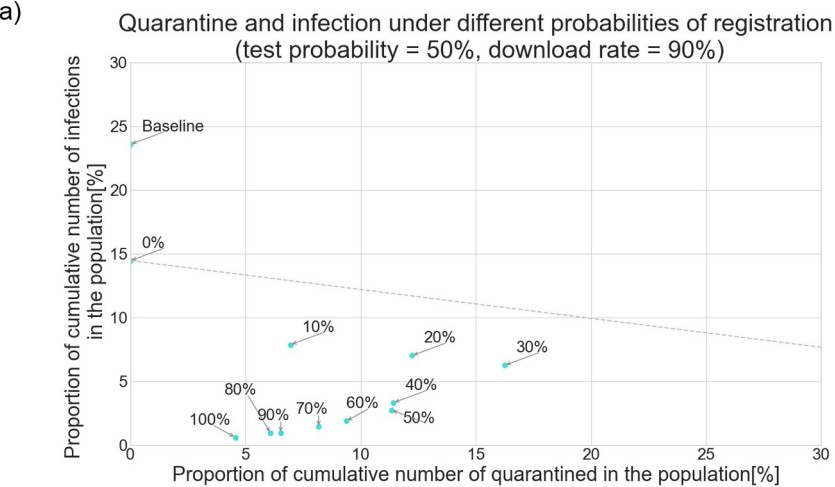

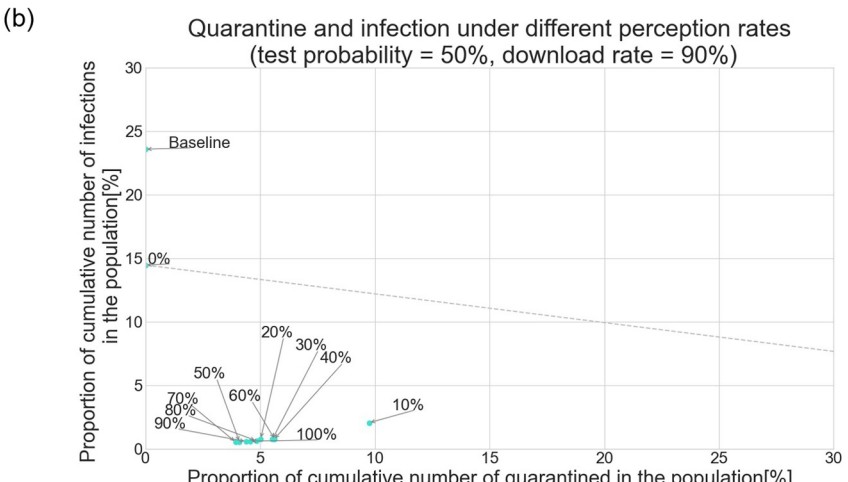

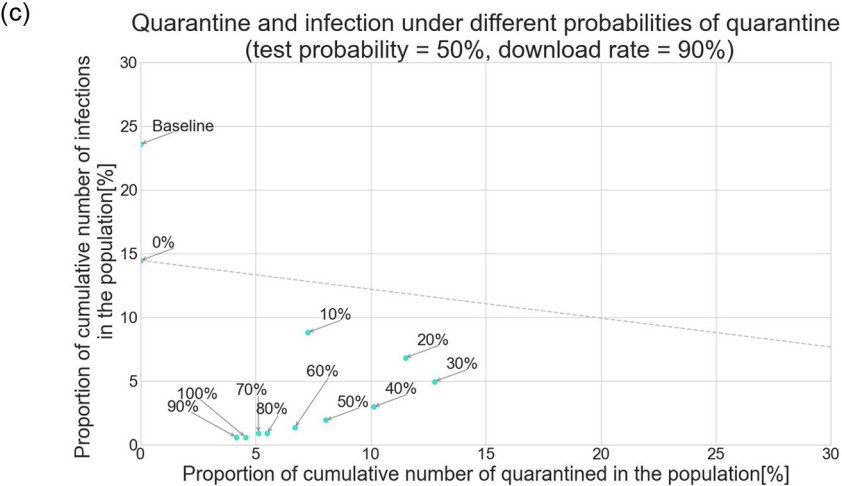

**Fig 6. Scenarios of imperfect registration of test results, perception rate, and quarantine.** (a) Scenario of varied probability of registration. (b) Scenario of varied perception rate. (c) Scenario of varied probability of quarantine.

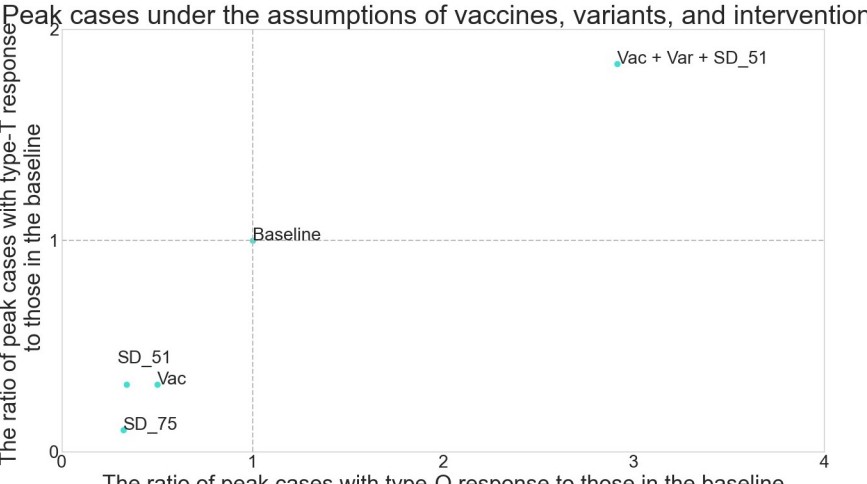

**Fig 7. The difference of peak cases considering vaccines, variants, and social distancing.**

transmissible variants account for a majority of cases. Moreover, social distancing measures such as restaurant closures and prohibition of large-scale events are still in place. These factors determine the speed of virus spread and thus could change the quantitative results regarding the difference between type-T and type-Q responses. This section discusses how the type-Q response's superiority to the type-T response changes when these assumptions are considered. Although mask-wearing is one of the most common interventions in reality, it is not considered because of the difficulty in quantifying its effects on transmissibility and susceptibility. The effectiveness of masks heavily depends on their type: cotton, surgical, or N95. There are no data available to detect the proportion of the users of each type of mask.

Fig 7 compares peak cases when type-T and type-Q responses are applied with vaccines, variants, and social distancing measures (Table 6), when the baseline results, where there are no interventions or new variants, are normalized to 1. The daily tests are assumed to be conducted on randomly selected symptomatic people with a probability of 50%, and the download rate is set to 90%. When social distancing is applied, 75% or 51% of the contacts that are randomly selected in the community and service-industry layers are negated, while their contacts in the home, workplace, school, and nursing home are maintained: Roughly, negating 75% and 51% of contacts correspond to the scenarios in which each person decreases their time outside the home by 50% or 30%, respectively, apart from commuting. Vaccines are assumed to be given to 0.7% of the population every day, with those aged over 65 years prioritized [26]. Those who are vaccinated with the first dose receive their second dose 21 days later. The

**Table 6. Scenarios to compare the difference between the two responses.**

| Scenario | Vaccines | Variants | Social-distancing |
|---|---|---|---|
| Baseline | No | No | No |
| Vac | Yes | No | No |
| SD_n | No | No | Yes (n% of all contacts in community and service-industry layers are negated.) |
| Vac + Var + SD_n | Yes | Yes | Yes (n% of all contacts in community and service-industry layers are negated.) |

susceptibility of those who have received the first and second doses decreases by 80% and 95%, respectively, compared to those who have not been vaccinated [27, 28]. New variants are also considered: they are 1.7 times as transmissible as the preexisting variants [29].

Results show that vaccines and social distancing substantially reduce peak cases when either response is applied. When vaccines are distributed or the number of people's contacts in the community and service-industry layers decreases by 51%, peak cases decrease by 50% to 70%; if such social distancing becomes stricter to decrease the contacts by 75%, peak cases decrease by more than 70% for each type of response. In each of these scenarios where only preexisting variants are assumed, peak cases decrease more with type-T response than with type-Q response, meaning that the relative superiority of the type-Q response decreases when compared to the baseline scenario, where no interventions are taken. This is because, even if the notified app users under the type-T response have contact with others, the chance of the virus spreading is rare because of low susceptibility thanks to vaccines or the small amount of contact from social distancing. When new variants are considered, transmissibility increases for both types of response. Here, the increase in peak cases with the type-Q response is 1.5 times larger than that with the type-T response, which means that the effectiveness of the type-Q response compared to that of the type-T response, again, decreases. This is because the new variant is so transmissible that even the type-Q response, which enables extreme curve-flattening in the baseline scenario, cannot sustain such effectiveness any longer. As a result, the spread of new variants makes the type-Q response less effective. In summary, when vaccines, variants, or social distancing are considered, there is less difference in peak cases under the two responses than in the baseline scenario, meaning the relative superiority of the type-Q response to the type-T response is weaker.

## 7 Conclusions, implications, and future research

By simulating the spread of COVID-19 using a detailed agent-based model and census data in Japan, this study provides a numerical analysis and projections on the effects of contact-tracing apps. Results show some crucial characteristics of the apps. First, apps' effectiveness depends on the intended response to a positive contact alert: The apps that indicate quarantine upon receiving a notification are more successful in achieving containment, whereas the apps that indicate testing have a limited curve-flattening effect. Second, the main factor contributing to the higher effectiveness of the former type is that noninfectious people whose positive status cannot be detected by current testing technology are quarantined. Third, if the download rate is extremely high, the apps with the type-Q response not only drastically reduce cases but also achieve containment with a small number of quarantined people. Finally, given the download rate of the apps, increasing the number of testing per day enhances the effectiveness of the apps, although the degree of enhancement is not proportional to the change in the amount of testing.

There are three possible directions for future research: First, the model can be extended to incorporate economic parameters. The evaluation of effectiveness in the current study is quite simple and hence leaves room for its development as a tool for economic analysis. If, for instance, social activities such as consumption and production are linked to people's interaction, one would be able to compute how much loss arises from isolation or quarantine. Second, another important intervention, namely the effectiveness of restricting inter-prefecture migration or movement across locations within a country, should be introduced. Finally, the parameters regarding people's contacts could be refined. For example, with data such as attributes of people with whom a person of a certain age tends to interact, the contacts in the model can be more similar to those in real life.

## Supporting information

**S1 Appendix.**
(PDF)

## Acknowledgments

I would like to express my very great appreciation to Keiichiro Kobayashi, Japan-SIR-Econ project members, the Cabinet Secretariat, and the Government Advisory Panel on COVID-19 for insightful comments from a practical point of view. I acknowledge data provision from the National Statistics Center.

## Author Contributions

**Conceptualization:** Asako Chiba.

**Data curation:** Asako Chiba.

**Formal analysis:** Asako Chiba.

**Funding acquisition:** Asako Chiba.

**Investigation:** Asako Chiba.

**Methodology:** Asako Chiba.

**Project administration:** Asako Chiba.

**Resources:** Asako Chiba.

**Software:** Asako Chiba.

**Supervision:** Asako Chiba.

**Validation:** Asako Chiba.

**Visualization:** Asako Chiba.

**Writing – original draft:** Asako Chiba.

**Writing – review & editing:** Asako Chiba.

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
