## [Decision Letter · Decision Letter 0]

16 Apr 2021

PONE-D-21-06447

Modeling the effects of contact-tracing apps on the spread of the coronavirus disease: mechanisms, conditions, and efficiency

PLOS ONE

Dear Dr. Chiba,

Thank you for submitting your manuscript to PLOS ONE. After careful consideration, we feel that it has merit but does not fully meet PLOS ONE’s publication criteria as it currently stands. Therefore, we invite you to submit a revised version of the manuscript that addresses the points raised during the review process.

The paper needs a MAJOR REVISION. Authors should follow the reviews in order to improve the quality of the paper.

We look forward to receiving your revised manuscript.

Kind regards,

Barbara Guidi

Academic Editor

PLOS ONE

Journal Requirements:

Reviewers' comments:

Reviewer's Responses to Questions

**Comments to the Author**

1. Is the manuscript technically sound, and do the data support the conclusions?

Reviewer #1: Yes

Reviewer #2: Yes

2. Has the statistical analysis been performed appropriately and rigorously? 

Reviewer #1: Yes

Reviewer #2: N/A

3. Have the authors made all data underlying the findings in their manuscript fully available?

Reviewer #1: No

Reviewer #2: No

4. Is the manuscript presented in an intelligible fashion and written in standard English?

Reviewer #1: Yes

Reviewer #2: Yes

5. Review Comments to the Author

Reviewer #1: This manuscript describes an agent-based model research design, intended to evaluate the effectiveness of contact tracing apps in mitigating the spread of COVID-19. They report that the apps are generally effective and, along with increased testing, the apps drastically reduced the number of cases reported. This project is within the scope of PLOS One, is well written, timely, and competently executed. I fully support acceptance for publication.

Reviewer #2: Study summary

The purpose of this study is to determine the effectiveness of various scenarios and responses to app-based contact-tracing in terms of the cumulative proportion of the given population who must isolate/quarantine, with the understanding that reducing the overall number of individuals who must isolate/quarantine will result in a reduction in the negative economic impact of the COVID-19 pandemic within that population. In particular, the author uses agent-based modeling to compare populations responding to contact-tracing apps by testing for COVID-19 after being informed that they have been in proximity to an infected individual and quarantining only if their test result is positive (a type-T response) vs. populations responding to the apps by immediately quarantining without first testing to determine whether or not they have been infected (a type-Q response). Type-Q scenarios therefore result in quarantining individuals who do not actually have the virus, in addition to those who have the virus but do not yet know it because they are pre-symptomatic and have not yet been tested and/or received their test results. Type-T scenarios result in fewer individuals being quarantined as a result of proximity to an individual with a confirmed infection, but also result in asymptomatic or pre-symptomatic individuals who have been infected by the primary patient circulating throughout the community and potentially spreading infection to new individuals. This author’s model is based largely off of the model (called Covasim) published by Kerr et al. (2020) and available open-source via either GitHub or the Python Package Index. The author has extended Kerr et al.’s Covasim model in several ways by including such things as isolation of severely ill or hospitalized individuals (reducing their contact with others and thus the possibility of passing illness to another), adding “super-spreader” individuals to the model [see this reviewer’s comments on that later in this review], and adding contacts among service workers and their customers, among other things. All of these characteristics are randomly generated and interact to simulate spread of COVID-19 in a population according to the parameters assigned to them by the model. Conditions in addition to the type-T and type-Q responses that are varied for comparison in the simulations are percent of symptomatic individuals tested per day (for example, 30%, 50%, 90%), and the percent of smartphone users who download and use the contact-tracing app.

The results of this study can potentially be used to inform policy decisions related to COVID-19.

How does this work differ from prior work? How does it add to the field?

Most (all?) other published studies modeling contact tracing apps assume a type-Q response. This study adds to our understanding of the usefulness of the apps by adding the type-T response to the model.

Other agent-based models are quite complex, thus limiting the ability to determine the main factors that generate the effects seen in their simulations. This agent-based model is simplified and includes only the variables that the author deems necessary to achieve a reasonably realistic simulation in order to be able to tease out the mechanisms underlying simulation results [see this reviewer’s comments on the model assumptions below].

Review

Overall I was impressed by the premise of the study, and feel that the use of agent-based modeling to reveal the most important factors affecting population-based spread of COVID-19 infections has great potential. The author’s focus on contact-tracing apps in this effort is very relevant, and the recognition of two distinct responses to contact-tracing alerts is astute and important in terms of producing simulations approximating real-world conditions. I greatly appreciate her approach. Additionally, the author’s analysis and description of the results of the study are very clear and complete. I enjoyed and appreciated reading that. However, there are also several areas where I feel the study could be improved, which I detail for the author below.

1. Assumptions of the model:

• Why did the author decide to add “superspreaders” to the model? This may well be a valid decision, but there is no explanation given. An explanation justifying this decision is important, as there are countless other conditions that the author also could have chosen to add to the model but did not. Why this one and not any of the others?

• Along the same lines, why does the author choose not to include the proportion of the population that is vaccinated? Would including it significantly change the differences you found between type-T populations and type-Q populations, and therefore change your conclusions?

• Is mask-wearing and social-distancing accounted for in the model? These are very common behaviors, and including these things would likely affect your results if they are not already accounted for. The contact-tracing app can only alert an individual if they were in proximity to an infected individual, not whether or not they were 5 feet apart vs. 7 feet, nor whether or not either was wearing a mask. A population that adheres to social-distancing and mask-wearing policies would most certainly reduce the proportion of type-T individuals who become infected and spread the virus before they are tested and receive results. Modeling this could significantly reduce the difference in the effects of type-T behaviors vs. type-Q behaviors. Adding this as a variable for comparison in your model might further increase its value in terms of a tool to inform policy decisions around the pandemic.

• The author may also want to include a way to vary presumed infectiousness of the viral variant, since this is an inevitable as long as the pandemic continues. Including this may also significantly change the differences you see between type-T and type-Q populations, and again might make your model more relevant.

2. Public access to your model:

I understand from your statement that the specific data from the Japanese census that you used to populate your model are not available publicly. However, is your model itself publicly accessible for people to populate with their own data? While it is not possible to run simulations using the exact dataset that you used, it would be possible to validate the model itself using a different dataset if it were publicaly available (for example, the Covasim model of Kerr et al.)

3. Related to point 2 above, it would be helpful for the author to make explicit in the paper that the Covasim model by Kerr et al. is in fact open-source. The link to the model should also be included, perhaps in a footnote.

4. Throughout the article, the author refers to the apps “requiring” a particular behavior (testing alone, or quarantine). The apps themselves do not require any particular behavior; they are simply a source of information, and it is the individual who determines their behavior. For this reason, a more precise orientation would be to refer to a “type-T response”, or a “type-Q behavior” as this reviewer has done in the summary section above.

5. I found Section 4.1 on comparison between SIR and ABM very interesting and informative. These are things that I do not have expertise in. However in the context of the author’s article, this discussion seemed out of place. SIR is not mentioned at any other point in the article, and comparison of the relative merits of SIR and ABR is not listed as an objective of the study. I believe that the use of ABR for your study is appropriate and stands on its own in the context of this work, and there is no need to describe or justify why this method was chosen instead of SIR. If you feel it belongs in the paper, I would suggest a short explanation in your conclusions section, or a note in your appendix.

Other suggested improvements:

1. Clarify imprecise or confusing terminology:

• For example, “superspreader” is not well-defined in its current usage. As the reference the author cites points out (Cave, 2020), “superspreader “is sometimes used to refer to an individual who may shed much more virus than most infected people and who is therefore more infectious, and it is sometimes used to refer to an event in which conditions are such that a large number of individuals become infected (a very crowded event with a long timeframe, for example). It was sometimes unclear what the author was referring to when the term was used. I ultimately determined that the author was referring to an individual, but this needs to be made clear from the start. Additionally, there is one place in the paper [p. 3, Section 2 (Model), line 4] where the author specifically uses the term “superspreading environments” which further adds to the confusion.

• “Lockdown” as it is used in the paper is another confusing term. I believe what the author means by this term is isolation or quarantine of an individual. However the common understanding of the term “lockdown” (in American culture at least) In addition, at one point [p. 1, line 2-3] the author uses the term and associates it with not only quarantine but also with behaviors such as social-distancing (in the US this term is specifically used to indicate individuals from different households staying at least 6 feet apart while in public).

2. Please add references in the following locations: p.2 line 2, following “… belong to the former category.”; p. 4, paragraph 2, line 2, after “… asymptomatic”.

3. p. 5, footnote 7: The author notes that in reality it takes 1-3 days to get test results. Why did then did the author choose to use 1 day in your model rather than using the mean (1.5 days)?

4. p. 11, line 2, after “…peak.” Add “(see fig. 3(c)). p. 11 line 3, what is meant by “subsidies”?

5. The author’s point on p. 11 would have more clarity and impact if you specify how many people the 0.3% represents in your model. Further, something like “If x% of the infected are severely ill the means x individuals would require hospitalization. For this model population there would be x hospital(s) with x beds. Thus in this scenario the hospitals would be overwhelmed” In other words, without some clear numbers to go with the author’s statements, they may easily sound like assertions without facts to back them up.

6. p. 12, figure 3, the x-axis does not represent “rates”. Consider labeling the x-axis to make it easier to interpret.

7. p. 19, paragraph 2, lines 10-11: Aren’t “isolation” and “quarantine” referring to the same thing here? If not please clarify.

8. p. 20, final sentence is very unclear. I am not sure what the author is trying to convey.

9. p. 23: Say a bit more about the effects/results under the other 2 conditions.

10. Font size in the figures is quite tiny, and I had difficulty reading it even with magnification. I suggest that the author increase it to 8 point. The author might consider shorter figure titles with a larger font. In some cases part of the figure title can be moved to an axis (see for example Figure 4).

11. The paper would benefit by use of a manuscript editing service, as there are places where wording and language are confusing, unclear, or incorrect.

To conclude, I would like to re-iterate how much I appreciate the topic and approach of this study. I really like the big idea behind it, and feel that it is both relevant and interesting. I thank the author for their very hard work, and I am grateful for the opportunity I was given to review this paper. I do hope that the author finds my comments useful, and also that this paper will be re-submitted following revisions.

6. PLOS authors have the option to publish the peer review history of their article (what does this mean?). If published, this will include your full peer review and any attached files.

Reviewer #1: **Yes: **Steven R. Holloway

Reviewer #2: **Yes: **Michelle L Mills

---

## [Author Response · Author response to Decision Letter 0]

24 May 2021

Response to Reviewer 1

Thank you very much for your supportive comments. Based on the other reviewer’s comments, I have revised the paper. Following the submission guideline, I used the template provided by PLOS. In addition, footnotes have been moved to the main text or the reference list. I believe the revision has improved the paper significantly.

Reviewer #1: This manuscript describes an agent-based model research design, intended to evaluate the effectiveness of contact tracing apps in mitigating the spread of COVID-19. They report that the apps are generally effective and, along with increased testing, the apps drastically reduced the number of cases reported. This project is within the scope of PLOS One, is well written, timely, and competently executed. I fully support acceptance for publication.

Response to Reviewer 2

Thank you very much for your insightful and detailed comments and suggestions. I believe they helped improve the paper significantly. Following the submission guideline, I used the template provided by PLOS. In addition, footnotes have been moved to the main text or the reference list. Below I have responded to each of your comments and detailed the relevant changes.

Review

Overall I was impressed by the premise of the study, and feel that the use of agent-based modeling to reveal the most important factors affecting population-based spread of COVID-19 infections has great potential. The author’s focus on contact-tracing apps in this effort is very relevant, and the recognition of two distinct responses to contact-tracing alerts is astute and important in terms of producing simulations approximating real-world conditions. I greatly appreciate her approach. Additionally, the author’s analysis and description of the results of the study are very clear and complete. I enjoyed and appreciated reading that. However, there are also several areas where I feel the study could be improved, which I detail for the author below.

1. Assumptions of the model:

• Why did the author decide to add “superspreaders” to the model? This may well be a valid decision, but there is no explanation given. An explanation justifying this decision is important, as there are countless other conditions that the author also could have chosen to add to the model but did not. Why this one and not any of the others?

Response: The reason for adding super-spreading environments is that the model should include the factors that determine the virus spread substantially. (Please note the expression “super spreader” was revised to “super-spreading environments.”)

For instance, the probabilities of infection, developing symptoms, and death in the model are varied across age with the interval of 10 years. A super-spreading environment is also a determinant of coronavirus spread. As described in Figure 2 in section 4.1 in the original paper, which was moved to Appendix B, the introduction of the super-spreading environments decreases peak cases by no less than 40%–50%. This means that the simulations overestimate cases if super-spreading environments are not considered.

Such rationale is added in section 2.3 in the revised paper.

• Along the same lines, why does the author choose not to include the proportion of the population that is vaccinated? Would including it significantly change the differences you found between type-T populations and type-Q populations, and therefore change your conclusions?

Response: Vaccines were not considered in the original analysis because they had not been widely distributed at the point of writing the first draft even though they are one of the factors that determines the speed of virus spread. At the same time, as mentioned in the following reviews, variants and interventions also change the quantitative results regarding the difference between type-T and type-Q responses. Therefore, section 6 was added to discuss these issues as follows. 

Figure 7 compares peak cases when type-T and type-Q responses are applied with vaccines, variants, and social distancing measures (Table 6), when the baseline results, where there are no interventions or new variants, are normalized to 1. The daily tests are assumed to be conducted on randomly selected symptomatic people with a probability of 50%, and the download rate is set to 90%. When social distancing is applied, 75% or 51% of the contacts which are randomly selected in the community and service-industry layers are negated, while their contacts in the home, workplace, school, and nursing home are maintained: Roughly, negating 75% and 51% of contacts correspond to the scenarios in which each person decreases their time outside the home by 50% or 30%, respectively, apart from commuting. Vaccines are assumed to be given to 0.7% of the population every day, with those aged over 65 years prioritized (Prime Minister’s Office of Japan, 2021). Those who are vaccinated with the first dose receive their second dose 21 days later. The susceptibility of those who have received the first and second doses decreases by 80% and 95%, respectively, compared to those who have not been vaccinated (Hunter and Brainard, 2021 and Polack et al., 2020). New variants are also considered: they are 1.7 times as transmissible as the preexisting variants (Davies et al., 2021).

Results show that vaccines and social distancing substantially reduce peak cases when either response is applied. When vaccines are distributed or the number of people’s contacts in the community and service-industry layers decreases by 51%, peak cases decrease by 50% to 70%; if such social distancing becomes stricter to decrease the contacts by 75%, peak cases decrease by more than 70% for each type of response. In each of these scenarios where only preexisting variants are assumed, peak cases decrease more with type-T response than with type-Q response, meaning that the relative superiority of the type-Q response decreases when compared to the baseline scenario, where no interventions are taken. This is because, even if the notified app users under the type-T response have contact with others, the chance of the virus spreading is rare because of low susceptibility thanks to vaccines or the small amount of contact from social distancing. When new variants are taken into account, transmissibility increases for both types of response. Here, the increase in peak cases with the type-Q response is 1.5 times larger than that with the type-T response, which means that the effectiveness of the type-Q response compared to that of the type-T response, again, decreases. This is because the new variant is so transmissible that even the type-Q response, which enables extreme curve-flattening in the baseline scenario, cannot sustain such effectiveness any longer. As a result, the spread of new variants makes the type-Q response less effective. In summary, when vaccines, variants, or social distancing are considered, there is less difference in peak cases under the two responses than in the baseline scenario, meaning the relative superiority of the type-Q response to the type-T response is weaker.

• Is mask-wearing and social-distancing accounted for in the model? These are very common behaviors, and including these things would likely affect your results if they are not already accounted for. The contact-tracing app can only alert an individual if they were in proximity to an infected individual, not whether or not they were 5 feet apart vs. 7 feet, nor whether or not either was wearing a mask. A population that adheres to social-distancing and mask-wearing policies would most certainly reduce the proportion of type-T individuals who become infected and spread the virus before they are tested and receive results. Modeling this could significantly reduce the difference in the effects of type-T behaviors vs. type-Q behaviors. Adding this as a variable for comparison in your model might further increase its value in terms of a tool to inform policy decisions around the pandemic.

Response: As pointed out, social distancing has two opposite effects on the results. On one hand, it suppresses virus spread because it reduces people’s contact; on the other hand, social distancing makes it difficult for the apps to detect actual contact, which hinders the effectiveness of the apps. As answered in the previous bullet point, the newly added simulations in section 6 incorporate only the former effect, considering that infections rarely occur between people more than 6 feet apart, the maximum distance many apps can detect. 

As for the analysis taking into account social distancing, please refer to the previous answer.

Although mask-wearing is one of the most common interventions, it is not included in this model because of the difficulty in quantifying its effects on transmissibility and susceptibility. The effectiveness of masks heavily depends on their type: cotton, surgical, or N95. There are no data available to detect the proportion of the users of each type of mask. This is mentioned in section 6.

• The author may also want to include a way to vary presumed infectiousness of the viral variant, since this is an inevitable as long as the pandemic continues. Including this may also significantly change the differences you see between type-T and type-Q populations, and again might make your model more relevant.

Response: Vaccines have been added to the analysis in section 6. As for the set-ups and results, please refer to the answer in the second bullet point.

2. Public access to your model:

I understand from your statement that the specific data from the Japanese census that you used to populate your model are not available publicly. However, is your model itself publicly accessible for people to populate with their own data? While it is not possible to run simulations using the exact dataset that you used, it would be possible to validate the model itself using a different dataset if it were publicaly available (for example, the Covasim model of Kerr et al.)

Response: Thank you very much for your suggestion. The Python codes used in the simulations has been uploaded in a zip file as Supporting Information.

3. Related to point 2 above, it would be helpful for the author to make explicit in the paper that the Covasim model by Kerr et al. is in fact open-source. The link to the model should also be included, perhaps in a footnote.

Response: In the third paragraph in section 1, I highlighted the fact that Covasim is an open-source tool. In addition, the outline of Covasim and the description of what is new in the present model is clarified in the first paragraph in section 2.

4. Throughout the article, the author refers to the apps “requiring” a particular behavior (testing alone, or quarantine). The apps themselves do not require any particular behavior; they are simply a source of information, and it is the individual who determines their behavior. For this reason, a more precise orientation would be to refer to a “type-T response”, or a “type-Q behavior” as this reviewer has done in the summary section above.

Response: Thank you for pointing out this imprecise language. The responses are referred to as “type-T/Q response” in the revised paper, and the phrasing “requiring” has been revised throughout.

5. I found Section 4.1 on comparison between SIR and ABM very interesting and informative. These are things that I do not have expertise in. However in the context of the author’s article, this discussion seemed out of place. SIR is not mentioned at any other point in the article, and comparison of the relative merits of SIR and ABR is not listed as an objective of the study. I believe that the use of ABR for your study is appropriate and stands on its own in the context of this work, and there is no need to describe or justify why this method was chosen instead of SIR. If you feel it belongs in the paper, I would suggest a short explanation in your conclusions section, or a note in your appendix.

Response: Thank you for your suggestion. Figure 2 in section 4.1 in the original paper was intended to explain how the assumption of heterogeneity in virus transmission, including super-spreading environments, changes the simulated speed of the outbreak. As the high-dimensional heterogeneity is the feature of this analysis, I chose to keep the figure. However, as you pointed out, the original expression, which frequently used the term “SIR” without its explanation, was inappropriate. Therefore, in the revised paper, the section was moved to appendix B, and a qualitative explanation for the analytical models was added as follows:

This section discusses how agent-based models are different from analytical epidemiological models, namely, susceptible-infected-removed (SIR) models (Anderson and May, 1979 and Kermack and McKendrick, 1927). In many analytical models, the number of newly infected people is determined by the product of the number of the infected and the number of the susceptible. The number of the severely ill and of the recovered are determined as a certain proportion of the infected. Thus, the overall mechanism of the transmission of virus in analytical models is equivalent to that in agent-based models, as presented in this paper. An obvious difference is that agent-based models take a bottom-up approach, meaning that the smallest unit in their structure is an individual, which enables analyses on inter-relationships among people. Fundamentally, this can be interpreted as high-dimensional heterogeneity. That is, in agent-based models, each individual is characterized by a variety of attributes and contacts. In other words, an agent-based model without this diversity is substantially the same as analytical epidemiological models.

Other suggested improvements:

1. Clarify imprecise or confusing terminology:

• For example, “superspreader” is not well-defined in its current usage. As the reference the author cites points out (Cave, 2020), “superspreader “is sometimes used to refer to an individual who may shed much more virus than most infected people and who is therefore more infectious, and it is sometimes used to refer to an event in which conditions are such that a large number of individuals become infected (a very crowded event with a long timeframe, for example). It was sometimes unclear what the author was referring to when the term was used. I ultimately determined that the author was referring to an individual, but this needs to be made clear from the start. Additionally, there is one place in the paper [p. 3, Section 2 (Model), line 4] where the author specifically uses the term “superspreading environments” which further adds to the confusion.

Response: Thank you for pointing out this confusion. The simulations assumed super-spreading environments, which refers to a phenomenon in which a small fraction of environments are high-risk compared to the total. The description and rationale behind this assumption are explained in section 2.3 in the revised paper, with some new citations, as follows:

A super-spreading environment refers to the phenomenon in which a small amount of cases accounts for a large amount of transmissions (Cave, 2020). Moreover, it has been argued that the occurrence of this phenomenon depends on the environment rather than on the infected individuals (Majra et al., 2020) and that the Pareto principle, namely that 80% of all consequences come from 20% of the causes, could apply in the case of COVID-19 super-spreading environments (Kumar et al., 2020). Therefore, it is reasonable to consider that transmission probability for 20% of contacts selected at random is 50 times as high as that for the remaining contacts in the simulations. In fact, super-spreading environments are one of the determinants of coronavirus spread. As described in Appendix B, introducing super-spreading environments into the model decreases peak cases by 40%-50%. This means that the simulations overestimate cases if super-spreading environments are not considered.

Accordingly, the term “super-spreader” was rephrased to “super-spreading environments.”

• “Lockdown” as it is used in the paper is another confusing term. I believe what the author means by this term is isolation or quarantine of an individual. However the common understanding of the term “lockdown” (in American culture at least) In addition, at one point [p. 1, line 2-3] the author uses the term and associates it with not only quarantine but also with behaviors such as social-distancing (in the US this term is specifically used to indicate individuals from different households staying at least 6 feet apart while in public).

Response: Thank you for your comment. “Lockdown” in the original text did not refer to social distancing but the prohibition of people’s going out or requirements for shops to close. To clarify the meaning, I rephrased the term “lockdown” to “stay-at-home order.”

2. Please add references in the following locations: p.2 line 2, following “… belong to the former category.”; p. 4, paragraph 2, line 2, after “… asymptomatic”.

Response: I have added the references to these two places as suggested.

3. p. 5, footnote 7: The author notes that in reality it takes 1-3 days to get test results. Why did then did the author choose to use 1 day in your model rather than using the mean (1.5 days)?

Response: The lag is set to one day in the analysis since it generally takes one day for tested individuals to receive results in Japan, with some cases of two or three days (Kittaka, 2020). The third paragraph in section 2.4 has been revised.

4. p. 11, line 2, after “…peak.” Add “(see fig. 3(c)). p. 11 line 3, what is meant by “subsidies”?

Response: “Subsidies” refers to the government’s financial aid for the private sector to compensate for the loss incurred by mandatory store closures in Japan. This part has been revised as follows:

Social losses would be heavy, such as a substantial increase in government subsidies to the private sector to compensate for losses incurred by mandatory store closures (Ministry of Economy, Trade and Industry, 2021) and economic stagnation. The amount of subsidies is high if the peak infection is high because the government then has to impose stringent and prolonged stay-at-home orders. In other words, a high peak not only harms people’s health but also requires large-scale government spending. Thus, the type-T response’s limitations in preventing virus spread are substantial.

5. The author’s point on p. 11 would have more clarity and impact if you specify how many people the 0.3% represents in your model. Further, something like “If x% of the infected are severely ill the means x individuals would require hospitalization. For this model population there would be x hospital(s) with x beds. Thus in this scenario the hospitals would be overwhelmed” In other words, without some clear numbers to go with the author’s statements, they may easily sound like assertions without facts to back them up.

Response: This part has been revised to describe the relationship between the results of the simulations and reality as follows:

Further, the peak value of the sum of severe and critical patients, who need hospitalization, accounts for 0.06% of the population. This would overwhelm hospitals, considering that the number of beds prepared for COVID-19 patients as of April in 2020 in Japan was 32,000, which is equivalent to 0.03% of the population (Ministry of Health, Labour and Welfare, 2021).

6. p. 12, figure 3, the x-axis does not represent “rates”. Consider labeling the x-axis to make it easier to interpret.

Response: Thank you. I labelled the x-axis to clarify that the figures are historical daily cases. 

7. p. 19, paragraph 2, lines 10-11: Aren’t “isolation” and “quarantine” referring to the same thing here? If not please clarify.

Response: Isolation refers to the behavior of those who are tested, find out they tested positive, and subsequently completely refrain from any type of contact until recovery, which describes hospitalization in reality. In contrast to isolation, quarantine refers to the behavior of the alerted app users in the case of type-Q response. People in quarantine reduce their daily contact with people outside their homes by 90%, while their contact with their family members, or others in nursing homes if applicable, would be as per usual.

Although paragraph 5 of section 2 in the original paper defined isolation and quarantine, it may have been unclear. Thus, the expressions have been slightly changed. In addition, section 2 is now divided into subsections to make it easier for readers to follow.

8. p. 20, final sentence is very unclear. I am not sure what the author is trying to convey.

Response: The sentence was revised to convey that the efficiency of apps is not always improved by increasing the test probability.

9. p. 23: Say a bit more about the effects/results under the other 2 conditions.

Response: Analysis regarding the two cases has been added, where the probability of registration and the probability of self-quarantine are varied, as follows:

To summarize, the number of the infected and thus the number of quarantined individuals are rarely affected by the apps’ perception rate because key contacts are detected regardless of the apps’ ability; when the probability of registration or of self-quarantine is less than 1, the number of effective app users is lower than the number of people who downloaded the app. Even if the download rate is as high as 90%, the small proportion of effective app users leads to the inefficiency problem observed in Figure 4. Thus, people’s honest registration of their test results and following self-quarantining are crucial for the apps to contribute to efficiently reducing infection numbers.

10. Font size in the figures is quite tiny, and I had difficulty reading it even with magnification. I suggest that the author increase it to 8 point. The author might consider shorter figure titles with a larger font. In some cases part of the figure title can be moved to an axis (see for example Figure 4).

Response: As suggested, the font size in every figure has been increased. For clarification, I chose not to shorten the titles of the figures.

11. The paper would benefit by use of a manuscript editing service, as there are places where wording and language are confusing, unclear, or incorrect.

Response: The manuscript has been reviewed by a professional English-editing service.

To conclude, I would like to re-iterate how much I appreciate the topic and approach of this study. I really like the big idea behind it, and feel that it is both relevant and interesting. I thank the author for their very hard work, and I am grateful for the opportunity I was given to review this paper. I do hope that the author finds my comments useful, and also that this paper will be re-submitted following revisions.

Again, thank you very much for your insightful and detailed comments and suggestions.

---

## [Decision Letter · Decision Letter 1]

1 Jul 2021

PONE-D-21-06447R1

Modeling the effects of contact-tracing apps on the spread of the coronavirus disease: mechanisms, conditions, and efficiency

PLOS ONE

Dear Dr. Chiba,

Thank you for submitting your manuscript to PLOS ONE. After careful consideration, we feel that it has merit but does not fully meet PLOS ONE’s publication criteria as it currently stands. Therefore, we invite you to submit a revised version of the manuscript that addresses the points raised during the review process.

The paper should be revised in order to address the MINOR REVISIONS suggested by the reviewers.

We look forward to receiving your revised manuscript.

Kind regards,

Barbara Guidi

Academic Editor

PLOS ONE

Journal Requirements:

Reviewers' comments:

Reviewer's Responses to Questions

**Comments to the Author**

1. If the authors have adequately addressed your comments raised in a previous round of review and you feel that this manuscript is now acceptable for publication, you may indicate that here to bypass the “Comments to the Author” section, enter your conflict of interest statement in the “Confidential to Editor” section, and submit your "Accept" recommendation.

Reviewer #2: (No Response)

2. Is the manuscript technically sound, and do the data support the conclusions?

Reviewer #2: Yes

3. Has the statistical analysis been performed appropriately and rigorously? 

Reviewer #2: Yes

4. Have the authors made all data underlying the findings in their manuscript fully available?

Reviewer #2: Yes

5. Is the manuscript presented in an intelligible fashion and written in standard English?

Reviewer #2: Yes

6. Review Comments to the Author

Reviewer #2: The author has done an excellent job addressing my comments in the initial review. I suggest the following minor improvements:

1.) Minor rewording and clarification of the abstract. Specifically, add a sentence just after the first sentence to briefly define the two responses to app alerts you are comparing (type-Q and type-T) and then edit the sentence beginning "First, with regard to contacts..." for English grammar and clarity.

2.) Page 2: line 55 - 56 --The end of the sentence for point 1. of your main findings is at the beginning of point 2.

3.) Page 4: line 133 -- The end of the sentence is missing.

4.) Page 4: line 135 - 136 -- Edit the first half of the first sentence under section 2.1 for English grammar and clarity.

5.) Page 5: Section 2.2, line 165 - 166 -- For the sentence beginning "They might become infectious..." does the author mean asymptomatic but infectious? using the term "asymptomatic infectious" rather than simply "infectious" would help better define and differentiate between "infectious" and "symptomatic", since a symptomatic individual is also infectious.

6.) Page 5: Line 178 -- add the year after "June 1 to 7".

Excellent paper. I look forward to seeing it in print.

7. PLOS authors have the option to publish the peer review history of their article (what does this mean?). If published, this will include your full peer review and any attached files.

Reviewer #2: No

---

## [Author Response · Author response to Decision Letter 1]

7 Jul 2021

Reviewer #2: The author has done an excellent job addressing my comments in the initial review. I suggest the following minor improvements:

I deeply appreciate your insightful comments that have enhanced the quality of the paper. I am glad to hear that the first round of revision satisfied the concerns raised in the previous review. Following are the responses to the concerns raised in this second round of review.

1.) Minor rewording and clarification of the abstract. Specifically, add a sentence just after the first sentence to briefly define the two responses to app alerts you are comparing (type-Q and type-T) and then edit the sentence beginning "First, with regard to contacts..." for English grammar and clarity.

Response: Thank you for your suggestion. I have revised the abstract so that the two types of responses are briefly defined and referred to.

2.) Page 2: line 55 - 56 --The end of the sentence for point 1. of your main findings is at the beginning of point 2.

Response: Point 1 and 2 have been revised.

3.) Page 4: line 133 -- The end of the sentence is missing.

Response: The codes are uploaded in protocols.io as suggested by the Editor. The sentence here has been revised to refer to the DOI of the uploaded file (http://dx.doi.org/10.17504/protocols.io.bwaepabe). The file is the same as the one submitted in response to the first-round review. Although the page in protocols.io is currently set as private, it is going to be published when the paper is published on the journal. The statement of no conflict of interest, originally inserted just after this sentence, has been removed because it was unnatural.

4.) Page 4: line 135 - 136 -- Edit the first half of the first sentence under section 2.1 for English grammar and clarity.

Response: Thank you for the advice. The sentence has been revised.

5.) Page 5: Section 2.2, line 165 - 166 -- For the sentence beginning "They might become infectious..." does the author mean asymptomatic but infectious? using the term "asymptomatic infectious" rather than simply "infectious" would help better define and differentiate between "infectious" and "symptomatic", since a symptomatic individual is also infectious.

Response: As pointed out, the term "infectious" in the sentence means asymptomatic but infectious. Therefore, it has been revised as suggested.

6.) Page 5: Line 178 -- add the year after "June 1 to 7".

Response: The year 2020 has been added for clarity and correctness.

---

## [Decision Letter · Decision Letter 2]

2 Aug 2021

Modeling the effects of contact-tracing apps on the spread of the coronavirus disease: mechanisms, conditions, and efficiency

PONE-D-21-06447R2

Dear Dr. Chiba,

We’re pleased to inform you that your manuscript has been judged scientifically suitable for publication and will be formally accepted for publication once it meets all outstanding technical requirements.

Kind regards,

Barbara Guidi

Academic Editor

PLOS ONE

Additional Editor Comments (optional):

Reviewers' comments:

Reviewer's Responses to Questions

**Comments to the Author**

1. If the authors have adequately addressed your comments raised in a previous round of review and you feel that this manuscript is now acceptable for publication, you may indicate that here to bypass the “Comments to the Author” section, enter your conflict of interest statement in the “Confidential to Editor” section, and submit your "Accept" recommendation.

Reviewer #2: All comments have been addressed

2. Is the manuscript technically sound, and do the data support the conclusions?

Reviewer #2: Yes

3. Has the statistical analysis been performed appropriately and rigorously? 

Reviewer #2: Yes

4. Have the authors made all data underlying the findings in their manuscript fully available?

Reviewer #2: Yes

5. Is the manuscript presented in an intelligible fashion and written in standard English?

Reviewer #2: Yes

6. Review Comments to the Author

Reviewer #2: This is excellent work. I look forward to seeing your paper published in a future edition of PLOS ONE.

7. PLOS authors have the option to publish the peer review history of their article (what does this mean?). If published, this will include your full peer review and any attached files.

Reviewer #2: **Yes: **Michelle Mills

---

## [Editor Report · Acceptance letter]

9 Aug 2021

PONE-D-21-06447R2 

Modeling the effects of contact-tracing apps on the spread of the coronavirus disease: mechanisms, conditions, and efficiency 

Dear Dr. Chiba:

I'm pleased to inform you that your manuscript has been deemed suitable for publication in PLOS ONE. Congratulations! Your manuscript is now with our production department. 

Kind regards, 

on behalf of

Dr. Barbara Guidi 

Academic Editor

PLOS ONE